# Prioritizing COVID-19 vaccine allocation in resource poor settings: Towards an Artificial Intelligence-enabled and Geospatial-assisted decision support framework

Soheil Shayegh[1‡], Javier Andreu-Perez[2,3☯], Caroline Akoth[4☯], Xavier Bosch-Capblanch[5,6☯], Shouro Dasgupta[7,8☯], Giacomo Falchetta[1,9☯], Simon Gregson[10,11☯], Ahmed T. Hammad[12,13☯], Mark Herringer[14,15☯], Festus Kapkea[4☯], Alvaro Labella[16☯], Luca Lisciotto[8,17☯], Luis Martínez[16☯], Peter M. Macharia[18,19,20☯], Paulina Morales-Ruiz[21☯], Njeri Murage[4☯], Vittoria Offeddu[22,23☯], Andy South[24☯], Aleksandra Torbica[25,26☯], Filippo Trentini[22,23,27☯], Alessia Melegaro[22,23,26‡]*

1 RFF-CMCC European Institute on Economics and the Environment, Centro Euro-Mediterraneo sui Cambiamenti Climatici, Milan, Italy, 2 Centre for Computational Intelligence, School of Computer Science and Electronic Engineering, University of Essex, Colchester, United Kingdom, 3 Group Simbad, Department of Computer Science, University of Jaén, Jaén, Spain, 4 Women in GIS Kenya Ltd, Nairobi, Kenya, 5 Swiss Tropical and Public Health Institute, Allschwil, Switzerland, 6 University of Basel, Basel, Switzerland, 7 Fondazione CMCC, Lecce, Italy, 8 Ca' Foscari University of Venice, Venice, Italy, 9 International Institute for Applied Systems Analysis, Vienna, Austria, 10 Imperial College School of Public Health, Imperial College London, London, United Kingdom, 11 Biomedical Research and Training Institute, Harare, Zimbabwe, 12 Università Cattolica del Sacro Cuore, Milan, Italy, 13 Decatab Pte. Ltd., Singapore, Singapore, 14 The Global Healthsites Mapping Project—Healthsites.io, Hoorn, Netherlands, 15 Mapping the Risk of International Infectious Disease Spread—mriids.org, Brookline, Massachusetts, United States of America, 16 Department of Computer Science, University of Jaén, Jaén, Spain, 17 DNV—Energy Systems, Bologna, Italy, 18 Department of Public Health, Institute of Tropical Medicine, Antwerp, Belgium, 19 Centre for Health Informatics, Computing and Statistics, Lancaster Medical School, Lancaster University, Lancaster, United Kingdom, 20 Population & Health Impact Surveillance GroupUnit, KEMRI-Wellcome Trust Research Programme, Nairobi, Kenya, 21 Faculty of Economics and Business, Access-to-Medicines Research Centre, Research Center for Operations Management, KU Leuven, Leuven, Belgium, 22 Covid Crisis Lab, Bocconi University, Milan, Italy, 23 Dondena Centre for Research on Social Dynamics and Public Policy, Bocconi University, Milan, Italy, 24 Liverpool School of Tropical Medicine, Liverpool, United Kingdom, 25 Cergas—Centre for Research on Health and Social Csare Management, SDA Bocconi School of Management, Bocconi University, Milan, Italy, 26 Department of Social and Political Science, Bocconi University, Milan, Italy, 27 Center for Health Emergencies, Bruno Kessler Foundation, Povo, Italy

☯ These authors contributed equally to this work.
‡ SS and AM are joint senior authors and contributed equally to this work
* alessia.melegaro@unibocconi.it

**Data Availability Statement:** The shapefiles and boundaries data come from this source (Macharia,

## Abstract

### Objectives

To propose a novel framework for COVID-19 vaccine allocation based on three components of Vulnerability, Vaccination, and Values (3Vs).

### Methods

A combination of geospatial data analysis and artificial intelligence methods for evaluating vulnerability factors at the local level and allocate vaccines according to a dynamic mechanism for updating vulnerability and vaccine uptake.

Peter M., Noel K. Joseph, and Emelda A. Okiro. "A
vulnerability index for COVID-19: spatial analysis at
the subnational level in Kenya." BMJ global health
5.8 (2020): e003014.) publicly available at https://
doi.org/10.6084/m9.figshare.12501455.v1 All
these data have a CC BY license and are provided
by Peter M Macharia who is the co-author of this
study.

**Funding:** AM and FT received funding from the
European Research Council (Project no.
101003183). VO and PMR received funding from
Fondazione Romeo and Enrica Invernizzi. PMM
was supported by the Royal Society Newton
Internal Fellowship (NIF/R1/201418). No other
authors received any funding for this research and
there was no additional external funding received
for this study.

**Competing interests:** The authors have declared
that no competing interests exist.

## Results

A novel approach is introduced including (I) Vulnerability data collection (including country-specific data on demographic, socioeconomic, epidemiological, healthcare, and environmental factors), (II) Vaccination prioritization through estimation of a unique Vulnerability Index composed of a range of factors selected and weighed through an Artificial Intelligence (AI-enabled) expert elicitation survey and scientific literature screening, and (III) Values consideration by identification of the most effective GIS-assisted allocation of vaccines at the local level, considering context-specific constraints and objectives.

## Conclusions

We showcase the performance of the 3Vs strategy by comparing it to the actual vaccination rollout in Kenya. We show that under the current strategy, socially vulnerable individuals comprise only 45% of all vaccinated people in Kenya while if the 3Vs strategy was implemented, this group would be the first to receive vaccines.

## Introduction

As COVID-19 mass vaccination campaigns in developed countries across Europe and Northern America are reaching a greater share of the population, many low- and middle-income countries (LMIC) in Asia, Africa, and Latin America are left behind facing growing challenges of securing the supply of vaccine, particularly, for their vulnerable populations to avoid the risk of new waves of infection [1]. Facing uncertainty, limited supply of vaccines, and the risk of growing vaccine hesitancy in relation to misinformation campaigns [2], these countries are forced to constantly revise their vaccination rollout plans to reflect realities on the ground and target the most vulnerable groups [3]. Meanwhile high-resolution spatial and temporal data that can be collected and shared through open licenses provide a unique opportunity to develop transparent and trustworthy Artificial Intelligence (AI)-supported tools and frameworks that can interoperate with existing health data platforms and integrate diverse expert opinions. Such tools could help public health decision-makers optimise their vaccine campaigns considering country-specific needs and geographical, institutional, and infrastructure constraints.

This paper introduces a framework for data collection and integration, evaluation of COVID-19 risks and allocation and prioritization of the available vaccine doses. The proposed framework combines knowledge from diverse disciplines, including geospatial, epidemiology, economics, statistics, and computational sciences. It employs spatial data analysis and visualization techniques to create an open platform for policymakers, community leaders, the scientific community, and other stakeholders to evaluate and compare different vaccination strategies based on principles and objectives outlined by the World Health Organization (WHO) Strategic Advisory Group of Experts on Immunization (SAGE) values framework for the allocation and prioritization of COVID-19 vaccination [4]. By focusing on Vulnerability, Vaccination, and Values (3Vs), this framework prioritizes vulnerable populations for receiving COVID-19 vaccine by considering the principles outlined in the WHO SAGE framework [5].

## Methods

The 3Vs framework is designed to help public health decision makers better allocate their resources and address vaccination inequalities. The 3Vs framework will be manifested in an

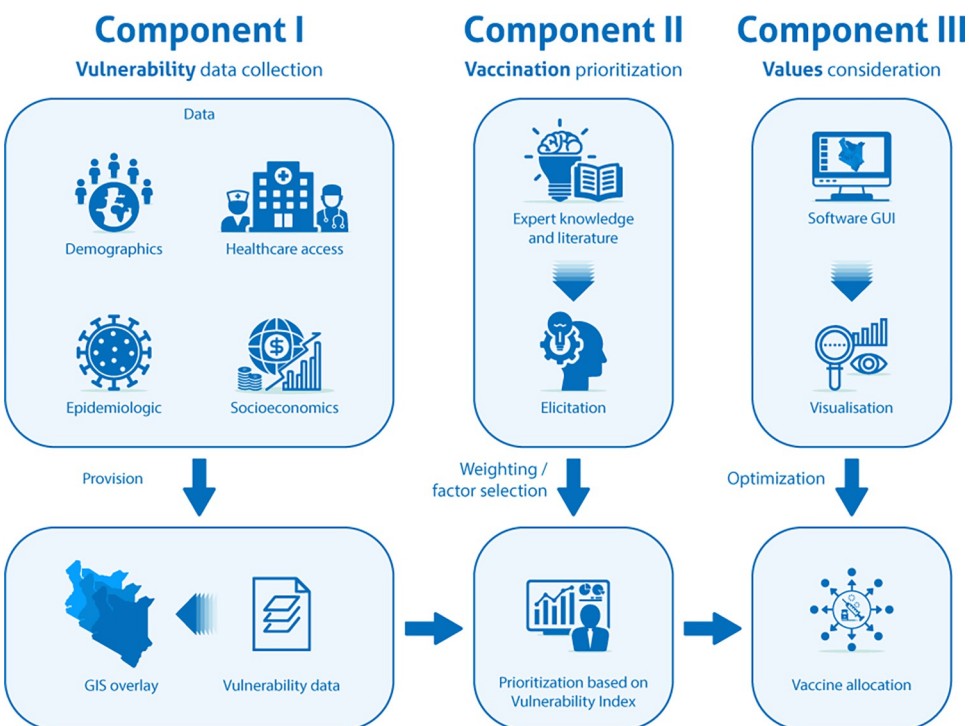

**Fig 1. The 3Vs framework.** Summary of the 3Vs (Vulnerability, Vaccination, and Values) framework and the relationships among its main components.

online interactive decision support platform aimed at optimizing COVID-19 country-specific vaccine rollout plans, from procurement to deployment and administration through the assessment and analysis of the following three components (Fig 1).

**Component I—Vulnerability data collection:** collection of data on socio-demographic, economic, epidemiological, and environmental characteristics of the affected communities, availability and accessibility of healthcare services, and other economic and societal indicators.

**Component II—Vaccination prioritization:** methods for compiling and translating data, with the help of AI-enabled expert elicitation techniques, into prioritization criteria spanning different health and demographic dimensions.

**Component III—Values consideration:** computational methods for geospatial vaccination allocation based on equity values enhanced by data visualization and communication tools.

A graphical representation of the 3Vs framework is provided in Fig 1.

The proposed framework will provide support to policy decision-makers through the availability and visualisation of vulnerable fragments of the population and their access to existing healthcare facilities at the finest geographical resolution. Furthermore, it will ensure that the factors considered in the first component are representative of most vulnerable populations including those belonging to historically underprivileged demographic and socioeconomic groups and, where necessary, identifying and including additional local elements. These may be related to the sustainability assessment of the vaccination program by evaluating its cost and benefits and its long-term impact on the healthcare system as well as other factors related to how individuals will respond to the vaccination program. Vaccine hesitancy might indeed further complicate any vaccination rollout plan. Through this approach several feasible options will be visualized and communicated to planners so that they will be able to make better and more informed decisions.

A multi-resolution and multiscale system model will be adopted so that different local priorities will be embedded in the algorithms and available to the users via an interactive online platform based on different layers of the spatial data. Stakeholders will be able to choose vulnerability factors among a set of predefined indicators and evaluate and compare the generated outputs.

## Component I–vulnerability data collection

The collection of the most up to date data from various heterogeneous sources will be used to construct a unique Vulnerability Index which will characterise the finest geographical units of the considered populations. Background data about the vulnerability of affected populations can be grouped into categories: socioeconomic, demographic, healthcare access, and epidemiological. As part of the activities within this component, a list of openly licensed geospatial information is retrieved from different sources and will be overlaid on one another. These include high-resolution population data [6,7], gridded gender and age distributions [7], degree of urbanization [8,9], travel friction surfaces [10], administrative boundaries and, where possible, health facility catchment areas [11], along with geotagged sample household survey information [12] on socio-demographics, health status, and cultural attitudes towards vaccines. Our framework also considers healthcare facilities location [13] and typologies [14,15] as well as their accessibility [16], and, where available, data on Intensive Care Unit (ICU) beds and vaccine hesitancy.

This baseline set of information, available (in part) for many LMICs, will provide the geospatial distribution of vulnerable population segments which drives the vaccine allocation plans. It will also paint a timely picture of the available healthcare facilities, along with critical logistics considerations such as the travel time for the population to reach those facilities. The gathered data will be complemented with region-specific information such as local or need-tailored data obtained from governments and development/aid agencies involved in the vaccination rollout planning and operationalization. Where available, local data on vaccine hesitancy will be incorporated into Component I. The success of COVID-19 vaccination as a pandemic mitigation strategy relies on its widespread acceptance [17], but early studies suggest that vaccine hesitancy could become a significant obstacle to COVID-19 vaccine uptake in many countries, including LMICs [18]. Integrating information on vaccine hesitancy into the framework would allow coordination of vaccination efforts with appropriate interventions to preempt or reduce refusal rates among target populations.

## Component II—vaccination prioritization

The ideal allocation of vaccines should consider a variety of factors that constitute the vulnerability of affected populations. In most countries, mass vaccination priorities have been driven, firstly, by targeting healthcare workers and frontline staff (e.g., teachers, security personnel) and, secondly, by age stratification [19,20], due to the age patterns of COVID-19 mortality [21]. However, in LMICs, the populations are much younger, and considering only the age factor, may overshadow other significant factors underlying the risk of severe infection or death. This is especially the case in communities with weak health infrastructure, with pre-existing health conditions, or without reliable access to healthcare services due to poverty or other socioeconomic conditions [22]. To identify the segments of the population that should be targeted by vaccination programs, the use of a multidimensional Vulnerability Index that reflects the underlying risk factors (demographic, health, and socioeconomic), is more nuanced than a unidimensional factor (e.g., age stratification). Early in the pandemic, existing indexes such as the Infection Diseases Vulnerability Index were used to assess the vulnerability and capacity to

respond to the pandemic at the country level in Africa [23,24] and other countries around the world [25,26].

**AI-enabled integration of expert knowledge for vulnerability index estimation.** Differently from existing direct data-driven models for estimating the Vulnerability Index, we used the suggestions of experts to decide which factors to include and their relative importance in defining the level of vulnerability in different geographical areas within a specific country. This allows to integrate the perceptual opinion of experts, as well as other key stakeholder groups to provide their insights and experience to bridge the knowledge gap between theoretical approaches to vulnerability assessment and the pandemic-related inequalities that many sub-populations are facing [27]. In our proposed approach, we have assembled and consulted a multidisciplinary panel of experts about their opinion on the following:

- A curated list of the relevant factors will be reviewed and updated based on more recent and available evidence and on the feedback received from a medical science, healthcare, and policy expert's panel.

- The relative importance of each factor will be considered by a weighting process that will be based on scientific, epidemiological, and health experts' opinions.

However, experts may have either convergent or divergent decisions which makes it necessary to implement AI-driven decision-making that convey that information in complete accord. Furthermore, these opinions can be subjective or vague and therefore an AI model should be able to account for the uncertainty of the provided responses. We suggest modelling the uncertain opinion of the experts via fuzzy linguistic variables that express information in terms of sets or intervals, rather than crisp numbers [28]. This way experts' opinions can be encoded into commonly used natural language expressions (viz. words) such as "Weakly important", "Fairly important", "Equally important", "Very important", with each expression denoting a possible fuzzy alignment to a determined rank level. Next there is the challenge of integrating divergent opinions and intelligently forming a consensus in a systematic way. We suggest using AI techniques for collective Multiple Criteria Decision Analysis (MCDA) such as derivatives approaches of the Best-Worst Method (BWM) [29] to compute the weight of each factor. Next, this later approach can be followed by a related Weighted Sum Method (WSM) [30] to calculate the unique index for each geographical division. These methods can account for the inconsistency of the surveying experts' opinions.

## Component III–values consideration

After building the Vulnerability Index using expert judgments, the proposed 3Vs framework builds on a mathematical allocation algorithm to assign available vaccines to designated vaccination centers considering vulnerability and accessibility, income/age/gender equity, and other feasibility considerations, including costs.

## Allocation method

The aim of the proposed strategy is to minimize the population-weighted average vulnerability across all regions (at the highest–available resolution). As a proof of concept, we set forth a stylized but effective recursive method that accounts for the relative priorities that determine the proportional size of the allocated number of doses to each area within a given country should be estimated based on the relatively vulnerable population in each area. In other words, the number of vaccines doses received by area $i$ (out of the $N$ areas) at time $t$ from the supplied

batch of vaccines with size $B_i^t$ will follow this equation:

$$x_i^t = \frac{P_i^{jt} \times V_i^t}{\sum_{n=1}^{N} P_n^t \times V_n^t} \times B^t \tag{1}$$

where $V_i^t$ is the Vulnerability Index and $P_i^t$ is the unvaccinated population of the area $i$ at the beginning of vaccination time $t$ with the vaccine batch $B^t$. Once the vaccines are distributed among vulnerable populations in each area, the number of unvaccinated people and the Vulnerability Index of each area should be updated accordingly for the next time step:

$$P_i^{t+1} = P_i^t - x_i^t \tag{2}$$

$$V_i^{t+1} = \frac{P_i^t \times V_i^t - x_i^t}{P_i^{t+1}} \tag{3}$$

With more individuals receiving a vaccine, the Vulnerability Index will decay towards a limit of $\hat{V}_i \geq 0$ which indicates the minimum feasible and acceptable vulnerability in the specific region $i$ due to vaccine hesitancy among the vulnerable population or logistical challenges in vaccinating all vulnerable populations.

## Results

To assess the validity of our proposed 3Vs framework, data from the recently developed Kenyan COVID-19 Social Vulnerability Index at the sub-county level was used [24]. These allowed the combination of 19 factors under three main categories related to socioeconomic inequality, population characteristics, and access to services. In addition, an array of microdata provided by the Demographic and Health Survey (DHS) was also included in our framework to characterise individuals' socio-demographic characteristics (age, employment, poverty, education) as well as their healthcare status (e.g., sanitation, vaccination). A detailed list of categories and the description of factors within each category together with the corresponding sources is presented in the Supporting information.

In Fig 2A the population distribution together with the healthcare facilities designated as main vaccination posts are shown (622 facilities out of the 13232 available in the country). Similarly, in Fig 2B we show the spatial distribution of the Social Vulnerability Index at the sub-county level [23] which is then used to perform the allocation of the vaccine doses following the procedure explained in the previous section. This allocation is based on population-weighted vulnerability and as a result, the hypothetical distribution of unvaccinated vulnerable population which derives from the implementation of the 3Vs framework shows very uniform and low vulnerabilities across the country (Fig 2C). On the other hand, mapping the current deployment of about 1 million vaccines in Kenya as of October 6, 2021, the vulnerability map of unvaccinated distribution does not change significantly and still closely mirrors the distribution of the population and the Social Vulnerability Index before vaccination (Fig 2D).

Comparing Fig 2A and 2B reveals that there may be some areas (such as the northern eastern parts of the country) with high vulnerability and doses' allocation requirements that may lack healthcare facilities to support vaccine distribution. On the contrary, a high number of healthcare facilities appear to be allocated in the western part of the country and around the capital (the blue circle in Fig 2A) where moderate social vulnerability and vaccination needs are derived from our analysis due to high concentration of economic opportunities and healthcare services.

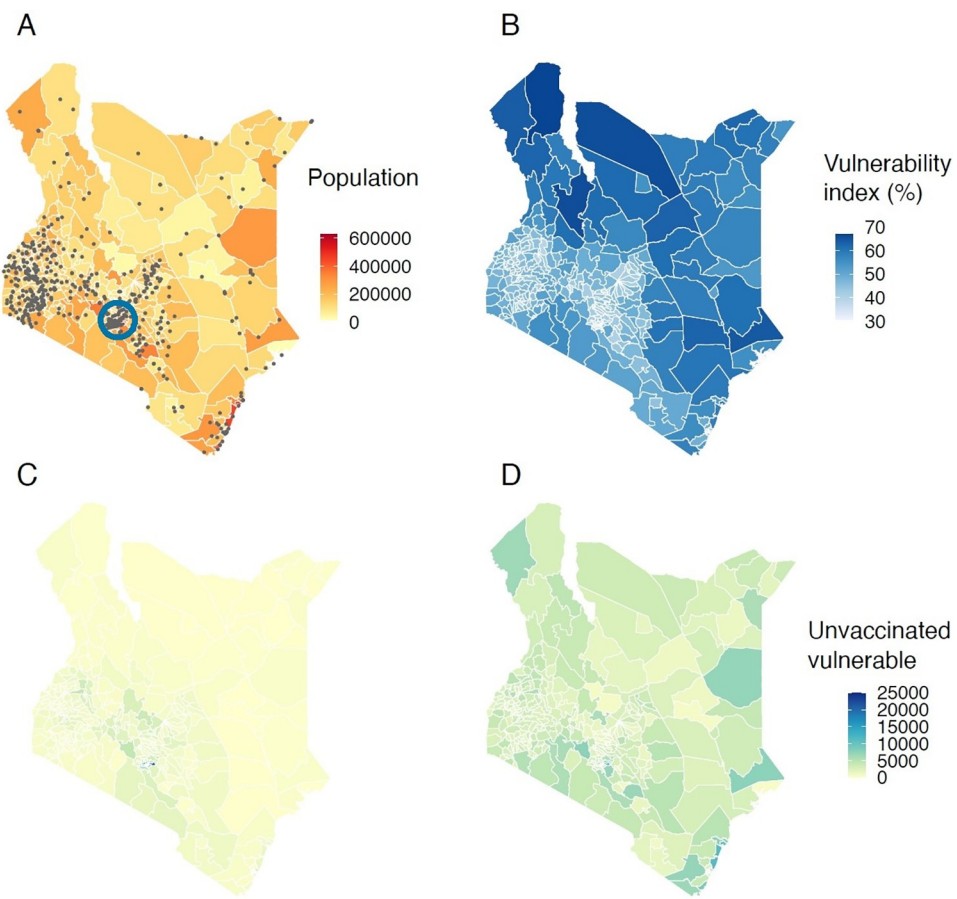

**Fig 2. Actual and proposed vaccine allocation distributions in Kenya.** (A) Population distribution at the sub-county level. The designated vaccination centres are shown with grey dots2 and the area around the capital, Nairobi, is identified with a blue circle. (B) Social Vulnerability Index distribution. (C) Distribution of unvaccinated vulnerable population after the implementation of the 3Vs strategy, and (D) distribution of unvaccinated vulnerable people after the implementation of current strategy.

To compare the performance of our proposed 3Vs strategy with the current vaccination rollout strategy in Kenya, we calculate the number of socially vulnerable populations in each sub-county who are left out of the vaccination program under each strategy. We estimate that under the current strategy, only about 450,000 vulnerable people have received vaccines and the rest of the vaccines have been administered to non-vulnerable populations. In other words, only about 45% of administered vaccines have gone to socially vulnerable groups. In comparison, if the proposed 3Vs strategy was used, the vaccines would be allocated only to vulnerable populations. It is worth mentioning that we do not consider vaccination hesitancy and outreach obstacles for this illustrative example and therefore, we assume all allocated vaccine doses would have been administrated. Fig 3A shows the spatial distribution of vulnerable people who would have received vaccines if our proposed strategy was implemented instead of the current strategy. Fig 3B in contrast, shows the benefits of the current strategy in terms of vulnerable populations unreached under our allocation scenario based on vulnerability. By summing up the numbers in each map and subtracting them, we find out that our strategy outperforms the current strategy by vaccinating an additional 550,000 vulnerable people in Kenya.

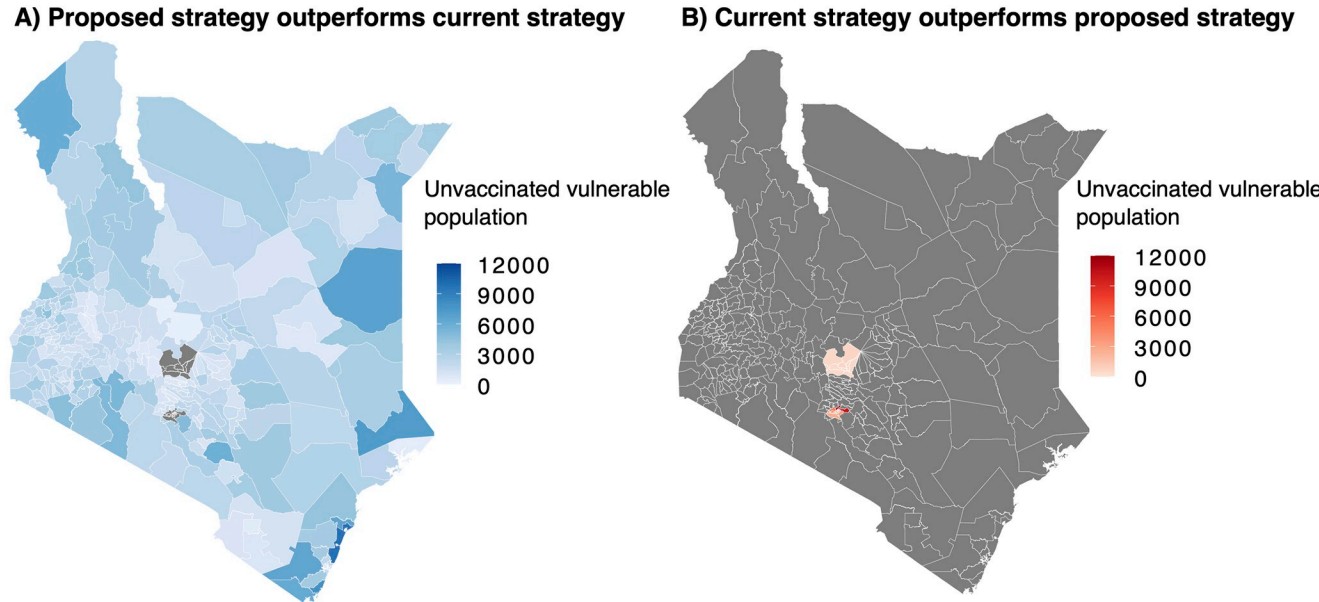

**Fig 3. Comparison of the performance of two strategies.** (A) distribution of unvaccinated and socially vulnerable people under the current strategy that could have been vaccinated if the proposed 3Vs strategy was adopted (B) distribution of socially vulnerable people that would have not been vaccinated under the proposed 3Vs strategy although they have been vaccinated under the current strategy.

## Discussion

The WHO SAGE framework outlines a series of principles that vaccination should follow and emphasizes on the value of fair access and equitable allocation in vaccine prioritization decisions. However, practical solutions at the national and local levels should be developed to support these overarching goals while guiding and enabling a well-informed decision-making process [4]. This situation is especially challenging in LMICs in sub-Saharan Africa, where several factors including underdeveloped and fragile healthcare systems, pervasive inequalities, widespread poverty and food insecurity, and high rates of infectious diseases can jeopardize the success of COVID-19 vaccination programs [31]. For example, shortage of healthcare facilities with reliable electricity supply necessary for cold chain and vaccine storage, and lack of physical access to these healthcare services within a reasonable travel time may hinder reaching vaccination targets and create further health inequalities between vaccinated and unvaccinated communities [32]. On the other side, there is evidence on much higher Covid-19 acceptance rates with respect to US or Russia [33] and it is crucial to translate these rates into actual uptake to prevent the risk of growing vaccine hesitancy in relation to targeted misinformation campaigns [34]. Therefore, our work aims at coupling goals and principles of vaccination strategies developed at the national level with more granular data about vulnerability and healthcare accessibility at the sub-national and local levels to support an informed decision-making process. In this position paper, we propose the 3Vs framework to support the recommendations and guidelines made by WHO for Covid-19 vaccine allocation. Our stylized example of vaccination rollout in Kenya shows the power of our approach in a clear way: while the proposed 3Vs strategy would distribute all vaccines to socially vulnerable populations across the country, under the current strategy more than half of the allocated doses have gone to people which are not considered socially vulnerable in our assessment.

Another important feature of our proposed strategy is the use of artificial intelligence. Given the compelling task of optimising the allocation of vaccines, the use of supportive

computational models, some of them AI-powered, have been a subject of consideration. Notwithstanding, the use of AI suggests some level of autonomy of the system. Therefore, we propose a limited, regulated, and transparent use of mathematical and statistical methods to achieve two specific challenges: 1) supporting experts' consensus through data-driven decision making; 2) estimating the relevancy of the risk factors that compose the vulnerability index. We reiterate that these AI-enabled tasks should not be performed fully autonomously, but in a human-in-the-loop fashion.

## Conclusions

By collecting and compiling data at the highest possible resolution, the 3Vs framework will produce a multidimensional Vulnerability Index not only for COVID-19 but for any other health emergency (such as climate change threats [35]). These indices will be critical in the decision-making process of allocating scarce resources, such as vaccines and healthcare services, in an efficient way to the most vulnerable segments of the populations. These tools can be integrated into already existing systems for decision-making at national or local levels in developing countries with limited access to high-quality data. Local non-governmental organizations and research communities in these countries should take the centre stage in developing such tools whose implementation and maintenance can be supported and funded by international aid and development organizations.

## Supporting information

**S1 Table. Vulnerability factors.**
(DOCX)

**S1 Appendix. Vulnerability data.**
(DOCX)

## Author Contributions

**Conceptualization:** Soheil Shayegh.

**Data curation:** Shouro Dasgupta.

**Investigation:** Soheil Shayegh.

**Methodology:** Soheil Shayegh.

**Supervision:** Alessia Melegaro.

**Visualization:** Javier Andreu-Perez.

**Writing – original draft:** Soheil Shayegh, Javier Andreu-Perez, Caroline Akoth, Xavier Bosch-Capblanch, Shouro Dasgupta, Giacomo Falchetta, Simon Gregson, Ahmed T. Hammad, Mark Herringer, Festus Kapkea, Alvaro Labella, Luca Lisciotto, Luis Martínez, Peter M. Macharia, Paulina Morales-Ruiz, Njeri Murage, Vittoria Offeddu, Andy South, Aleksandra Torbica, Filippo Trentini, Alessia Melegaro.

**Writing – review & editing:** Soheil Shayegh, Javier Andreu-Perez, Caroline Akoth, Xavier Bosch-Capblanch, Shouro Dasgupta, Giacomo Falchetta, Simon Gregson, Ahmed T. Hammad, Mark Herringer, Festus Kapkea, Alvaro Labella, Luca Lisciotto, Luis Martínez, Peter M. Macharia, Paulina Morales-Ruiz, Njeri Murage, Vittoria Offeddu, Andy South, Aleksandra Torbica, Filippo Trentini, Alessia Melegaro.

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
