## [Decision Letter · Decision Letter 0]

18 Jul 2023

PONE-D-22-22930Prioritizing COVID-19 vaccine allocation in resource poor settings:

Towards an Artificial Intelligence-enabled and Geospatial-assisted decision support frameworkPLOS ONE

Dear Dr. Melegaro,

Thank you for submitting your manuscript to PLOS ONE. After careful consideration, we feel that it has merit to be published in the PLOS ONE. There is a minor comment to restructure the Abstract of the manuscript following the PLOS ONE format. Therefore, we invite you to submit a revised version of the manuscript that addresses the points raised during the review process.

We look forward to receiving your revised manuscript.

Kind regards,

Chandan Kumar, Ph.D.

Academic Editor

PLOS ONE

Journal Requirements:

“AM and FT received funding from the European Research Council (Project no. 101003183). VO and PMR received funding from Fondazione Romeo and Enrica Invernizzi. PMM was supported by the Royal Society Newton Internal Fellowship (NIF/R1/201418).”

3. We note that Figures 2 & 3 in your submission contain [map/satellite] images which may be copyrighted. All PLOS content is published under the Creative Commons Attribution License (CC BY 4.0), which means that the manuscript, images, and Supporting Information files will be freely available online, and any third party is permitted to access, download, copy, distribute, and use these materials in any way, even commercially, with proper attribution. For these reasons, we cannot publish previously copyrighted maps or satellite images created using proprietary data, such as Google software (Google Maps, Street View, and Earth). For more information, see our copyright guidelines: http://journals.plos.org/plosone/s/licenses-and-copyright.

a. You may seek permission from the original copyright holder of Figures 2 & 3 to publish the content specifically under the CC BY 4.0 license. 

Additional Editor Comments :

Although I understand this article cannot be strictly structured similar to any empirical study, however, the authors have used the empirical structure (quantitative study) to design this manuscript very well. Similarly, the abstract can also be restructured to follow the PLOS ONE format. Authors may even use their own classifications, such as:

• Context

• Proposition

• Data source

• Evidence

• Conclusions

Reviewers' comments:

Reviewer's Responses to Questions

**Comments to the Author**

1. Is the manuscript technically sound, and do the data support the conclusions?

Reviewer #1: Yes

2. Has the statistical analysis been performed appropriately and rigorously? 

Reviewer #1: Yes

3. Have the authors made all data underlying the findings in their manuscript fully available?

Reviewer #1: Yes

4. Is the manuscript presented in an intelligible fashion and written in standard English?

Reviewer #1: Yes

5. Review Comments to the Author

Reviewer #1: The authors have presented an important issue regarding COVID-19 vaccine allocation in resource poor settings using an AI and GIS assisted decision support framework. This framework will be useful for other developing countries.

6. PLOS authors have the option to publish the peer review history of their article (what does this mean?). If published, this will include your full peer review and any attached files.

Reviewer #1: **Yes: **Denny John

---

## [Author Response · Author response to Decision Letter 0]

26 Jul 2023

We have addressed all points that were raised. In particular the following actions have been undertaken:

1- PLOS ONE's style requirements.

We have revised the manuscript and named the files according to PLOS ONE’s style.

2- Funding Statement.

We have updated the funding statement and included all missing information.

“AM and FT received funding from the European Research Council (Project no. 101003183). VO and PMR received funding from Fondazione Romeo and Enrica Invernizzi. PMM was supported by the Royal Society Newton Internal Fellowship (NIF/R1/201418). No other authors received any funding for this research and there was no additional external funding received for this study.”

3- Copyrighted images.

We confirm that all images used in the manuscript are original images generated from publicly accessible data. The shapefiles and boundaries data come from this source (Macharia, Peter M., Noel K. Joseph, and Emelda A. Okiro. "A vulnerability index for COVID-19: spatial analysis at the subnational level in Kenya." BMJ global health 5.8 (2020): e003014.) publicly available at this link: https://doi.org/10.6084/m9.figshare.12501455.v1

All these data have a CC BY license and are provided by Peter M Macharia who is the co-author of this study.

4- Abstract.

We have restructured the abstract following your suggestion.

---

## [Editor Report · Decision Letter 1]

27 Jul 2023

Prioritizing COVID-19 vaccine allocation in resource poor settings:

Towards an Artificial Intelligence-enabled and Geospatial-assisted decision support framework

PONE-D-22-22930R1

Dear Dr. Melegaro,

We’re pleased to inform you that your manuscript has been judged scientifically suitable for publication and will be formally accepted for publication once it meets all outstanding technical requirements.

Kind regards,

Chandan Kumar, Ph.D.

Academic Editor

PLOS ONE

---

## [Editor Report · Acceptance letter]

31 Jul 2023

PONE-D-22-22930R1 

Prioritizing COVID-19 vaccine allocation in resource poor settings:
Towards an Artificial Intelligence-enabled and Geospatial-assisted decision support framework 

Dear Dr. Melegaro:

I'm pleased to inform you that your manuscript has been deemed suitable for publication in PLOS ONE. Congratulations! Your manuscript is now with our production department. 

Kind regards, 

on behalf of

Dr. Chandan Kumar 

Academic Editor

PLOS ONE